# Serum Beta-Secretase 1 Activity Is a Potential Marker for the Differential Diagnosis between Alzheimer’s Disease and Frontotemporal Dementia: A Pilot Study

**DOI:** 10.3390/ijms25158354

**Published:** 2024-07-30

**Authors:** Claudia Saraceno, Carlo Cervellati, Alessandro Trentini, Daniela Crescenti, Antonio Longobardi, Andrea Geviti, Natale Salvatore Bonfiglio, Sonia Bellini, Roland Nicsanu, Silvia Fostinelli, Gianmarco Mola, Raffaella Riccetti, Davide Vito Moretti, Orazio Zanetti, Giuliano Binetti, Giovanni Zuliani, Roberta Ghidoni

**Affiliations:** 1Molecular Markers Laboratory, IRCCS Istituto Centro San Giovanni di Dio Fatebenefratelli, 25125 Brescia, Italy; csaraceno@fatebenefratelli.eu (C.S.); dcrescenti@fatebenefratelli.eu (D.C.); alongobardi@fatebenefratelli.eu (A.L.); sbellini@fatebenefratelli.eu (S.B.); r.nicsanu@campus.unimib.it (R.N.); 2Department of Translational Medicine and for Romagna, University of Ferrara, 44121 Ferrara, Italy; crvcrl@unife.it (C.C.); gianmarco.mola@edu.unife.it (G.M.); zlngnn@unife.it (G.Z.); 3Department of Environmental and Prevention Sciences, University of Ferrara, 44121 Ferrara, Italy; trnlsn@unife.it (A.T.); raffaella.riccetti@edu.unife.it (R.R.); 4Service of Statistics, IRCCS Istituto Centro San Giovanni di Dio Fatebenefratelli, 25125 Brescia, Italy; ageviti@fatebenefratelli.eu (A.G.); nbonfiglio@fatebenefratelli.eu (N.S.B.); 5MAC–Memory Clinic and Molecular Markers Laboratory, IRCCS Istituto Centro San Giovanni di Dio Fatebenefratelli, 25125 Brescia, Italy; sfostinelli@fatebenefratelli.eu (S.F.); gbinetti@fatebenefratelli.eu (G.B.); 6Alzheimer’s Rehabilitation Operative Unit, IRCCS Istituto Centro San Giovanni di Dio Fatebenefratelli, 25125 Brescia, Italy; dmoretti@fatebenefratelli.eu; 7Alzheimer’s Research Unit, IRCCS Istituto Centro San Giovanni di Dio Fatebenefratelli, 25125 Brescia, Italy; ozanetti@fatebenefratelli.eu

**Keywords:** BACE1, Alzheimer’s disease, frontotemporal dementia, GFAP, NfL, serum, biomarker, differential diagnosis

## Abstract

Alzheimer’s disease (AD) and frontotemporal dementia (FTD) are the two major neurodegenerative diseases causing dementia. Due to similar clinical phenotypes, differential diagnosis is challenging without specific biomarkers. Beta-site Amyloid Precursor Protein cleaving enzyme 1 (BACE1) is a β-secretase pivotal in AD pathogenesis. In AD and mild cognitive impairment subjects, BACE1 activity is increased in brain/cerebrospinal fluid, and plasma levels appear to reflect those in the brain. In this study, we aim to evaluate serum BACE1 activity in FTD, since, to date, there is no evidence about its role. The serum of 30 FTD patients and 30 controls was analyzed to evaluate (i) BACE1 activity, using a fluorescent assay, and (ii) Glial Fibrillary Acid Protein (GFAP) and Neurofilament Light chain (NfL) levels, using a Simoa kit. As expected, a significant increase in GFAP and NfL levels was observed in FTD patients compared to controls. Serum BACE1 activity was not altered in FTD patients. A significant increase in serum BACE1 activity was shown in AD vs. FTD and controls. Our results support the hypothesis that serum BACE1 activity is a potential biomarker for the differential diagnosis between AD and FTD.

## 1. Introduction

Alzheimer’s disease (AD) is by far the most common cause of dementia (60–70% of cases), affecting around 55 million people in the elderly population worldwide, with 10 million new cases each year [1,2,3,4]. On the contrary, only 5% of dementia cases seem to be caused by frontotemporal dementia (FTD) [5], which, however, represents the second most common form of early-onset dementia with clinical presentations in individuals under 65 years old [6,7]. FTD involves frontal and temporal brain region degeneration and is marked by abnormalities in personality, language, and executive function [8,9]. It is known that AD and FTD are the major neurodegenerative diseases with distinct clinical and neuropathological profiles that ultimately result in dementia [10]. The differential diagnosis between these neurodegenerative disorders is challenging without specific biomarkers. Thus, the identification of a biomarker to discriminate between these pathologies is essential.

Beta-site Amyloid Precursor Protein (APP) cleaving enzyme 1 (BACE1) has been deeply investigated as a promising biomarker for the diagnosis of AD [11,12]. BACE1 is a β-secretase, a key enzyme in the formation of amyloid-β (Aβ), pivotal in AD pathogenesis, catalyzing the rate-limiting initial cleavage at the β site of APP [13,14,15]. For this reason, BACE1 has been widely studied as part of brain amyloidogenesis and proven to be involved in Aβ production based on data from different knockout mouse models [16,17,18]. High BACE1 activity was found in the human AD brain, according to the findings that neurons produce the highest Aβ levels [19,20]. During the last 20 years, several human in vivo studies have shown a good diagnostic performance of cerebrospinal fluid (CSF) BACE1 activity/levels in discriminating AD patients from mild cognitive impairment (MCI) subjects and healthy controls [21,22,23,24,25,26]. BACE1 activity is increased in both brain/CSF of AD patients [27] and MCI subjects [22,28]. Moreover, it is also present in plasma [29,30] and in platelets [31,32,33], where its levels appear to reflect those in the brain [28]. Consistently, several studies have shown an increase in plasma/serum BACE1 activity in AD patients and MCI converting to AD (MCI-AD) [29,34,35]. The association of plasma BACE1 activity with CSF biomarkers of dementia, tau protein and Aβ42 peptide [36,37], has been previously demonstrated [29]. Notably, our recent study has shown a significant correlation of BACE1 activity with levels of Aβ40, Aβ42, and Aβ40/42 ratio in serum [38]. Moreover, we have found that serum BACE1 activity was able to discriminate AD and MCI-AD from controls with high sensitivity and specificity (98% and 100%, respectively) [38]. All of these results support the idea of plasma/serum BACE1 activity as an early biomarker of AD.

To date, there is no evidence of BACE1 ability to discriminate AD from other types of dementia, such as FTD. Thus, the relationship between BACE1 and FTD is still to be known. Recently, higher levels of Glial Fibrillary Acid Protein (GFAP) and Neurofilament Light chain (NfL) in the serum of FTD and AD patients compared to healthy controls were reported [39,40,41,42]. GFAP concentration in FTD was associated with disease severity and disability and correlated with deficits in cognitive domains. These notions support the hypothesis of serum GFAP as a marker of disease intensity and severity in FTD patients [39]. Serum NfL concentrations showed high accuracy in identifying FTD patients from cognitively healthy elderly subjects, correlating with cognition and GABAergic deficits. Thus, it is considered a promising biomarker of disease severity in FTD [40]. Moreover, in a study on a well-documented UK cohort, GFAP and NfL levels were evaluated in a dementia-free population after adjustment for multiple confounders that may affect protein expression levels (e.g., age and sex). Elevated peripheral levels of both proteins were associated with cognitive impairment and dementia, being able to distinguish participants with dementia [43]. GFAP is a marker of astrogliosis that is increased in the brains [44] and CSF of AD patients [45,46], making it a potential biomarker candidate for AD. However, reactive astrogliosis in the brain is also a marker of FTD [47]. This is why increased levels of GFAP are also observed in FTD patients. Elevated blood GFAP levels were described as a marker of ongoing astrogliosis in AD patients [41]. Moreover, different studies showed a strong association of peripheral GFAP levels with amyloid pathology [48,49,50,51]. All of these studies reporting non-specific changes in blood GFAP and NfL levels as markers of early neuroinflammation and neuronal damage in different types of dementia, including AD and FTD, promote the research of a new marker able to discriminate between these pathologies. In this pilot study, we used a small-sized cohort of patients to evaluate the role of BACE1 in FTD. In particular, we investigated whether serum BACE1 activity is altered in FTD patients.

## 2. Results

### 2.1. BACE1 Activity Is Not Altered in FTD Patients

BACE1 activity and GFAP and NfL levels were evaluated in the serum of 30 FTD patients and 30 subjects with normal cognitive function (CTRL). No differences were found in BACE1 activity between the two groups (mean ± SEM; FTD, 12.96 ± 0.92 kU/L vs. CTRL, 14.30 ± 1.14 kU/L, Generalized Linear Model, *p* > 0.05) (Figure 1a). A significant increase in GFAP and NfL levels was observed in FTD patients compared to CTRL subjects (mean ± SEM; GFAP: FTD, 152.60 ± 16.32 pg/mL vs. CTRL, 75.55 ± 13.26 pg/mL, Generalized Linear Model, *p* = 0.0178; NfL: FTD, 25.28 ± 2.54 pg/mL vs. CTRL, 10.30 ± 0.88 pg/mL, Generalized Linear Model, *p* < 0.0001) (Figure 1b,c). All comparisons were adjusted for age and gender. Age was significantly and positively associated with both GFAP and NfL levels.

### 2.2. BACE1 Activity Is Altered in AD Compared to FTD Patients

We compared serum BACE1 activity in FTD and AD patients. To this aim, we included in the analysis serum BACE1 activity data previously obtained by our group in 31 AD patients and 30 CTRL [13]. A significant increase in BACE1 activity was shown in AD patients compared to CTRL and FTD patients (mean ± SEM; AD, 20.88 ± 3.41 kU/L vs. CTRL, 10.97 ± 0.73 kU/L, Generalized Linear Model, p_adj_ < 0.001; AD vs. FTD, 12.96 ± 0.92 kU/L, Generalized Linear Model, p_adj_ = 0.007). No differences were observed between FTD and CTRL (p_adj_ = 0.12) (Figure 2a). All comparisons were adjusted for age and gender. Age was significantly and negatively associated with BACE1. Then, we evaluated GFAP and NfL levels. We observed a significant increase in GFAP levels in both AD and FTD patients compared to CTRL (mean ± SEM; AD, 236.70 ± 15.81 pg/mL vs. CTRL, 100.90 ± 9.42 pg/mL, Generalized Linear Model, p_adj_ < 0.001; FTD, 152.60 ± 16.32 pg/mL vs. CTRL, Generalized Linear Model, p_adj_ = 0.002) and in AD compared to FTD patients (AD vs. FTD, Generalized Linear Model, p_adj_ = 0.012) (Figure 2b). Moreover, we observed a significant increase in NfL levels in AD and FTD patients compared to CTRL (mean ± SEM; AD, 21.70 ± 1.88 pg/mL vs. CTRL, 12.83 ± 0.87 pg/mL, Generalized Linear Model, p_adj_ < 0.001; FTD, 25.28 ± 2.54 pg/mL vs. CTRL, Generalized Linear Model, p_adj_ < 0.001). No differences were observed between FTD and AD patients (*p* > 0.05) (Figure 2c). All comparisons were adjusted with Bonferroni post hoc correction and for age and gender as confounders. Age was significantly and positively associated with both GFAP and NfL levels. Gender was associated with GFAP, with males having lower levels of GFAP than females.

### 2.3. BACE1 Activity Discriminates AD from FTD Patients

In order to evaluate the capacity of serum BACE1 activity and GFAP levels to classify subjects as AD or FTD patients, a classification tree (CT) was implemented. NfL was not included in the CT since there was no difference between AD and FTD patients. The model had an overall correct classification performance of 83.6%. While GFAP was able to perform the first discrimination between AD and FTD patients, BACE1 activity had an important role in discriminating subjects with GFAP levels higher than 143.97 pg/mL. Among those, subjects with a serum BACE1 activity higher than 15.32 kU/L were classified as AD patients with a very high percentage (95.7%) (Figure 3).

## 3. Discussion

AD and FTD are the two major neurodegenerative diseases causing dementia [10]. Due to their similar clinical phenotypes, such as memory disturbances [52] and behavioral abnormalities [53], differential diagnosis between these pathologies is still challenging without specific biomarkers. Therefore, it is essential to identify a specific biomarker for an early and accurate differential diagnosis. This potential biomarker should be cost-effective and easy to perform in order to avoid invasive approaches (e.g., CSF withdrawal) and/or the use of expensive technology (neuroimaging devices).

We devoted our attention to serum BACE1 activity for a number of reasons. First, the mRNA expression, protein concentration, and enzymatic activity of this protein are higher in the brain of AD and MCI compared to cognitively healthy subjects [22,27,28]. Second, as previously demonstrated by our group and other researchers, alterations in brain BACE1 levels reflect changes not only in CSF but also in plasma/serum [22,27,28,29]. Third, the employed assay has excellent analytical performance (low intra- and inter-assay variability) and is affordable, easy to perform, and suitable for everyday clinical practice. Fourth, in a previous large study, we found that serum BACE1 activity was able to discriminate AD from a group defined as “other dementia”, including Lewy Body disease, Parkinson’s dementia, FTD, and similar diseases [54].

In the present study, we show that serum BACE1 activity could be a candidate biomarker for discriminating AD from FTD. Indeed, we found that this biomarker is selectively increased in AD patients (already shown in [38]), with higher levels compared to both FTD and cognitively healthy subjects. Moreover, the significant increase in GFAP and NfL levels observed in FTD and AD patients compared to controls confirms the presence of neuronal damage and corroborates previous data regarding the capability of these biomarkers in identifying FTD, AD, and related dementia compared to the cognitively healthy elderly, as well as in assessing FTD and AD severity and prognosis [39,40,55,56]. In this scenario, BACE1 activity plays a key role in discriminating subjects with GFAP levels higher than 143.97 pg/mL. In particular, among those, subjects with serum BACE1 activity higher than 15.32 kU/L were identified as AD patients, with a high percentage of 95.7%. In addition, no correlation between serum BACE1 activity and serum GFAP and NfL levels was observed in FTD patients, suggesting that neural damage and astrogliosis may not influence the peripheral levels of β-secretase. Several studies have contributed to the identification of non-invasive blood biomarkers for the differential diagnosis of neurodegenerative dementias, such as FTD and AD, whose clinical symptoms and pathological features frequently overlap. Among these biomarkers, GFAP and NfL are considered important biomarkers of AD and FTD pathogenesis but non-specific. GFAP is the most important cytoskeletal component of astrocytes. Reactive astrocytosis, shown by elevated peripheral GFAP levels, has been recognized as a potential starting point of AD. NfL is a component of myelinated axons and is released under neuronal damage [57,58]. Elevated blood GFAP and NfL levels can be observed in other neurodegenerative diseases (e.g., Parkinson’s disease, amyotrophic lateral sclerosis), stroke, or traumatic brain injury [56,57]. Thus, their non-specificity for dementia limits further clinical applications. Even if the capacity of blood GFAP and NfL levels to differentiate AD from FTD has already been investigated, with variable findings, to date, p-Tau181 is more suitable in the differential diagnosis of these disorders, showing its high diagnostic value [39,40,42,55,56,59,60,61]. These results encourage the search for a new promising biomarker. To the best of our knowledge, this is the first study testing serum BACE1 activity as a tool for the differential diagnosis between AD and FTD. This enzyme is deeply involved in AD pathogenesis since it plays a prominent role in Aβ homeostasis and is the main determinant of the Aβ42 isoform [13,15]. Recent studies point to the role of BACE1 in the degradation of Aβ42 in the less toxic and more soluble isoforms of these peptides [62,63]. According to the accumulation of neurotoxic Aβ that originates before the onset of AD symptoms, we recently demonstrated an increase in BACE1 activity in the serum of MCI-AD patients. This result demonstrates the high capability of serum BACE1 activity to discriminate subjects with a high probability of developing AD at the early stages of the pathology [38]. For all of these reasons, BACE1 could be a potential target for therapeutic approaches. Unfortunately, some compounds assessing its disease-modifying capacity in phase II/III clinical trials were discontinued due to futility or toxicity problems [64,65,66,67].

This study has both limitations and strengths that must be acknowledged. First, the cross-sectional nature of the investigation prevents us from drawing any conclusive considerations about the cause–effect relationship between the variables of interest. Second, the sample size is relatively small, although it was justified by a power analysis. Therefore, a replication study on a larger independent cohort is warranted to confirm the findings. Third, we cannot exclude that biases or unmeasured confounding factors might limit the robustness of our findings. Specifically, we cannot rule out that undetected brain changes, including established biomarkers for the differential diagnosis of AD, such as atrophy and the appearance of cerebral lacunes [68], as well as those holding promise, such as variations in intracranial compliance, brain creep, and glymphatic system activity [69,70], may have influenced our results. These limitations should be considered in future studies.

We would also like to highlight the strengths of the present study. We are the first to show that serum BACE1 activity could be a candidate biomarker for distinguishing AD from FTD. This finding is valuable, considering some important features of the BACE1 detection method: excellent analytical performance (low intra- and inter-assay variability), affordability, and easy execution. All of these characteristics make this assay suitable for any standard biochemistry laboratory.

In conclusion, these findings suggest that serum BACE1 activity may not only serve as an early biomarker for AD, aiding in the identification of eligible participants for clinical trials, but also as a potential marker to differentiate AD from FTD patients.

## 4. Materials and Methods

### 4.1. Subjects

In this retrospective study, we included 121 subjects, *n* = 30 FTD, *n* = 31 AD, and *n* = 60 CTRL. Among these subjects, *n* = 30 CTRL and *n* = 31 AD were characterized in [38]. Samples were obtained from the biobank of IRCCS Fatebenefratelli, Brescia. Subjects considered for this study underwent clinical and neurological examination at the MAC-Memory Clinic IRCCS Fatebenefratelli, Brescia. A clinical diagnosis was made according to international guidelines [71,72,73,74]. Demographic characteristics are reported in Table 1.

### 4.2. Biochemical Analyses

The serum BACE1 activity assay was performed by the Department of Translational Medicine and for Romagna from the University of Ferrara as described in [34,38]. Briefly, the BACE1 substrate was the peptide SEVNLDAEFR labeled with the fluorescent group Lucifer Yellow and with the quenching group Dabsyl (see details in [34]). The substrate was dissolved in dimethyl sulfoxide at a concentration of 392 μM and stored in aliquots at −20 °C for up to 3 months (the basal signal was stable within this storage time). For the assay, the stock solution was diluted to a final concentration of 30 μM in 50 mM acetate buffer, pH 4.5, and 0.1 M NaCl. One hundred microliters of this substrate solution at a final concentration of 30 μM were dispensed in triplicate in the wells of a black, flat-bottom microplate. Following a pre-incubation period of 10 min at 37 °C, the reaction was started by adding 5 µL of undiluted serum, and the fluorescence was read every 30 s for 20 min using excitation and emission wavelengths of 430 nm and 520 nm in a Tecan Infinite M200 (Tecan Group, Männedorf, Switzerland) microplate reader. The reaction rates were converted from relative fluorescence units (RFU) per minute to enzyme units (U) by interpolation with a standard curve constructed using known concentrations of the wild-type enzyme (β-secretase human; Sigma-Aldrich, Saint Louis, MO, USA, Cat. No. S4195).

Levels of GFAP and NfL were measured simultaneously in the serum of 30 FTD, 31 AD, and 60 CTRL using the commercially available Simoa Human Neurology 2-Plex B (N2PB) assay kit (Quanterix, Lexington, MA, USA) on the automated Simoa SR-X analyzer (Quanterix, Lexington, MA, USA), following the manufacturer’s instructions. Briefly, samples were thawed at room temperature for 60 min and centrifuged at 10,000× *g* for 5 min prior to analyses, as suggested in the protocol, to prevent any sample debris from interfering with the measurement. One hundred microliters of calibrators were added in triplicate in appropriate wells. Twenty-five microliters of controls provided in the kit or samples were dispensed in duplicate after a 1:4 dilution, adding 75 µL of sample diluent. Then, 20 µL of vortexed beads were added to each well, followed by 20 µL of detector reagent. After incubation at 35 °C for 30 min, the plate with pelleted beads was transferred onto the Simoa SR-X instrument for the analyses.

### 4.3. Statistical Analysis

The sample size was calculated by a power analysis with BACE1 activity as the primary outcome, considering as a reference the measurements performed in [38]. Since we showed serum BACE1 activity of 7.63 ± 1.88 kU/L for the CTRL group and 19.39 ± 9.66 kU/L for the AD + MCI-AD group (mean ± standard deviation, two-tailed Mann–Whitney non-parametric test, alpha level = 0.05, power of 0.9), we calculated that 10 subjects per group was the minimum sample size necessary for this variation to be significant.

The normality assumption of continuous variables was assessed by the Shapiro–Wilk test and graphical inspection. We employed the chi-square test to evaluate the relationship between categorical variables and the group variable. For the non-normally distributed demographic variable age, we used the Kruskal–Wallis test along with Dunn’s multiple comparison test. To examine the significant differences in biomarker levels among groups, we applied Generalized Linear Models, adjusting for age and gender. When comparing more than two groups (AD vs. FTD vs. CTRL), Bonferroni post hoc adjustment was implemented, and the relative adjusted *p*-values were reported. All statistical tests were two-tailed, with statistical significance set at *p* < 0.05. A classification tree (CT) [75] was applied to determine a profile of AD and FTD patients based on a set of biomarkers. Classification trees are a nonlinear modeling technique that recursively splits data based on features to create subsets, making decisions at each node. Even if features are correlated, the tree will choose the one providing the best information gain for splitting. In detail, CT model was carried out, considering the AD and FTD diagnoses as the two classes, depending on serum BACE1 activity and serum GFAP levels as quantitative independent variables. The output of the CT consists of a tree-like structure formed by different classification pathways (defined by the best discriminating estimated cut-offs of the biomarkers). At each split, the probability of the most likely diagnostic group based on the biomarker levels was provided. All analyses were conducted using Rstudio (R version: 4.3.2), except for the CT, which was built with SPSS (v.29).

### 4.4. Ethics Committee

All subjects (or legal guardians) provided written informed consent. To ensure that the appropriate ethical standards were upheld (respect of persons, beneficence, and justice), the study protocol was reviewed and approved by the local ethics committee (Prot. N. 91/2019, 57/2022; date of approval: 4 December 2019, 8 November 2022).

## Figures and Tables

**Figure 1 ijms-25-08354-f001:**
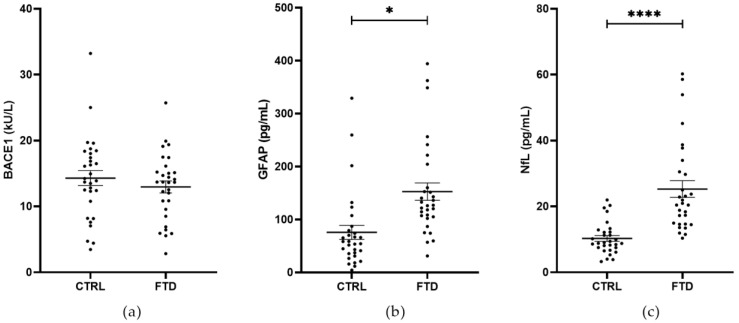
BACE1 activity (**a**) and the levels of GFAP (**b**) and NfL (**c**) in the serum of CTRL (*n* = 30) and FTD patients (*n* = 30). No differences were observed in BACE1 activity between the two groups. A significant increase in GFAP and NfL levels was shown in FTD patients compared to CTRL. Mean ± SEM; * *p* < 0.05 and **** *p* < 0.0001.

**Figure 2 ijms-25-08354-f002:**
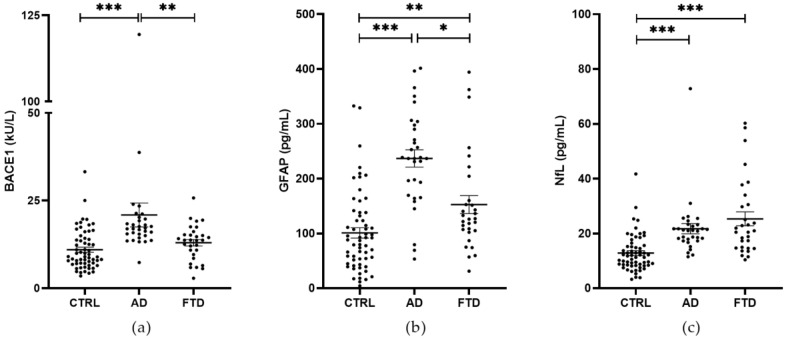
BACE1 activity (**a**) and the levels of GFAP (**b**) and NfL (**c**) in the serum of CTRL (*n* = 60), AD (*n* = 31), and FTD (*n* = 30) patients. A significant increase in BACE1 activity was shown in AD patients compared to CTRL and FTD patients. No differences were observed between CTRL and FTD patients. A significant increase in both GFAP and NfL levels was observed in AD and FTD patients compared to CTRL. Moreover, a significant increase in GFAP levels was shown in AD compared to FTD patients. No differences of NfL levels were observed between AD and FTD patients. Mean ± SEM; * *p* < 0.05, ** *p* < 0.01, and *** *p* < 0.001.

**Figure 3 ijms-25-08354-f003:**
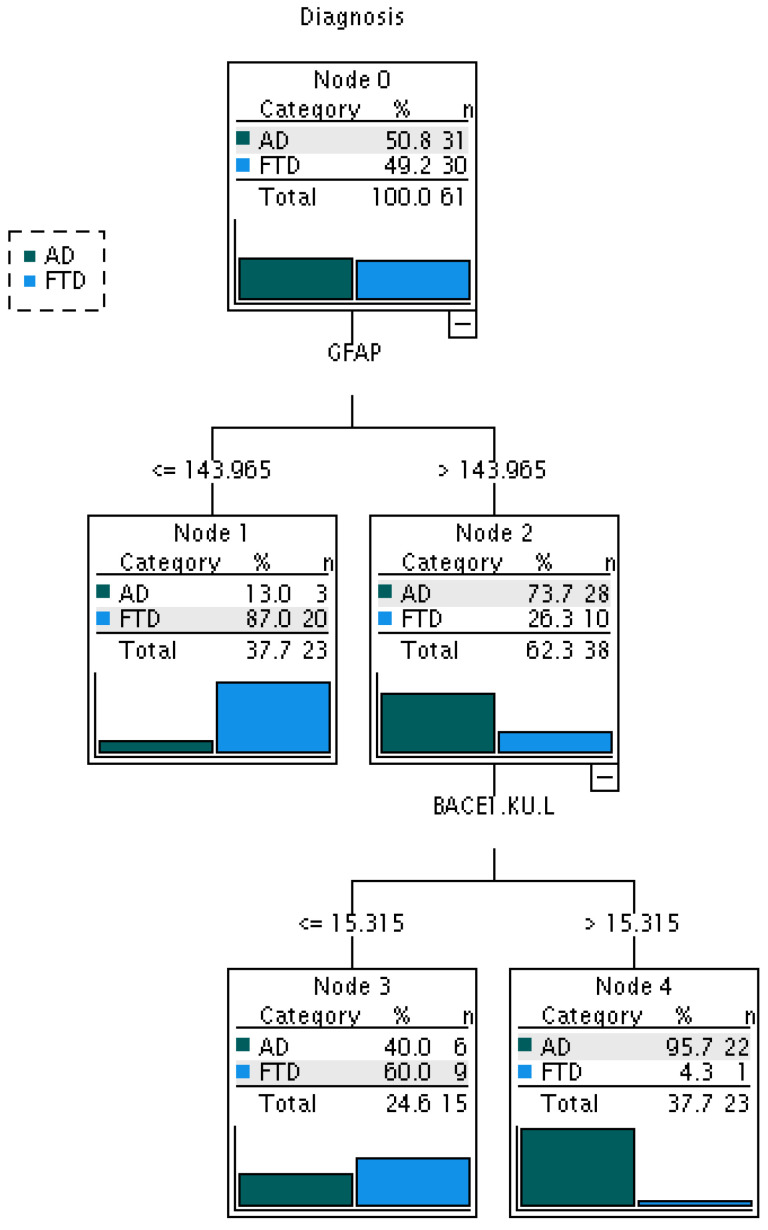
Classification tree obtained for AD (*n* = 31) vs. FTD (*n* = 30) patients based on BACE1 activity and GFAP levels in serum. BACE1 activity resulted in being able to discriminate AD from FTD patients with very high percentage (95.7%).

**Table 1 ijms-25-08354-t001:** The demographic characteristics of patients and controls included in this study.

	CTRL	AD	FTD	*p*-Value
N.	60	31	30	
Sex (% female)	65.00	67.74	33.33	0.0071 ^a^
Age, years	68.97 ± 5.99	69.16 ± 10.65	72.97 ± 7.67	0.0606 ^b^

CTRL, control; AD, Alzheimer’s disease; FTD, frontotemporal dementia. ^a^ chi-square test; ^b^ Kruskal–Wallis test. Means ± standard deviation.

## Data Availability

The data presented in this study are available in the Zenodo Data Repository at the following doi: 10.5281/zenodo.11145962 [76] (dataset creation date: 8 May 2024).

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
