# Peer review of "Serum Beta-Secretase 1 Activity Is a Potential Marker for the Differential Diagnosis between Alzheimer’s Disease and Frontotemporal Dementia: A Pilot Study"

_ijms, 2024, doi:10.3390/ijms25158354_

Round 1

Reviewer 1 Report (Previous Reviewer 2)

Comments and Suggestions for Authors

1. The abstract section does not include the numerical findings of this paper.

2. the sample size of the study is relatively small (n=30 for each group), which raises concerns about the statistical power and generalizability of the findings. I am not optimistic about accepting this paper because a larger cohort is necessary to validate the results and ensure they are not due to random variation.

3. The study is cross-sectional, limiting its ability to establish causal relationships between serum BACE1 activity and the progression of Alzheimer’s Disease (AD) or Frontotemporal Dementia (FTD). Longitudinal studies are essential to confirm the utility of BACE1 as a biomarker over time.

4. The study does not adequately address potential confounding variables that could influence BACE1 activity, such as comorbidities, medication use, and lifestyle factors. The lack of comprehensive control measures undermines the reliability of the findings. So, these findings are not relieble enough for clinical users.

5. The study does not provide a thorough comparison of serum BACE1 activity with established biomarkers such as tau protein, amyloid-beta levels, or neuroimaging data. This comparison is crucial to demonstrate the added value of BACE1 as a differential diagnostic tool.

6. The methodology section lacks sufficient detail regarding the assay validation, including specificity, sensitivity, and reproducibility of the BACE1 activity measurements. Moreover, the potential influence of pre-analytical factors (e.g., sample handling, storage conditions) on BACE1 activity is not discussed.

7. The study primarily uses generalized linear models (GLM) without providing sufficient justification for their selection over other potentially more appropriate statistical methods. The choice of GLM should be explained in the context of the data distribution and research questions.

8. The analysis does not adequately adjust for multiple potential confounders beyond age and gender. Factors such as comorbidities, medication use, genetic predispositions (e.g., APOE genotype), and lifestyle factors (e.g., diet, physical activity) should be considered in multivariable models to ensure robust results.

9. The detailed information for multivariate analysis is unavailable, such as the type of possible corrections and dependency analysis of the features.

10. The classification tree (CT) analysis used to differentiate AD from FTD patients lacks validation:

10.1.The authors should implement cross-validation techniques, such as k-fold cross-validation or bootstrapping, to assess the stability and generalizability of the CT model.

10.2. The study does not include ROC analysis to evaluate the diagnostic performance of serum BACE1 activity. ROC curves and corresponding area under the curve (AUC) values should be presented to demonstrate the sensitivity and specificity of BACE1 activity in distinguishing between AD and FTD.

10.3. Other metrics for generalizability analysis should also be reported for your results.

11. Although a power analysis is mentioned, the details are insufficiently reported. The manuscript should include a comprehensive power calculation, clearly outlining the assumptions, effect sizes, and expected variance to justify the sample size.

12. The study compares multiple biomarkers across groups but does not adequately address the issue of multiple comparisons, which increases the risk of Type I errors. Appropriate corrections, such as Bonferroni correction or False Discovery Rate (FDR) adjustment, should be applied and reported.

13. The robustness of the findings is not tested through sensitivity analyses. The authors should conduct and report sensitivity analyses to examine how changes in key assumptions or parameters affect the study outcomes.

14. The selection of control subjects is not well-explained, and there may be biases in how controls were chosen compared to the patient groups. Random selection methods and matching techniques (e.g., propensity score matching) should be employed to minimize selection bias.

15. The study findings have not been externally validated in an independent cohort, which is a critical step for confirming the robustness and applicability of the results in different populations.

16. The clinical applicability of measuring serum BACE1 activity as a routine diagnostic test remains unclear. The authors should provide more evidence on the feasibility, cost-effectiveness, and practical implementation of this biomarker in clinical settings.

17. Although the authors acknowledge some limitations, there is insufficient discussion on the potential impact of these limitations on the study’s conclusions.

18. The future trend section needs to be completed. Previous studies confirm the importance of intracranial compliance as a biomarker for brain disorders such as hydrocephalus. You can suggest evaluating intracranial compliance as a marker to differentiate between Alzheimer’s Disease and Frontotemporal Dementia in future studies.

Comments on the Quality of English Language

Moderate level

Author Response

Reviewer 2 Report (Previous Reviewer 3)

Comments and Suggestions for Authors

The previous concerns have been addressed.

Author Response

Thank you very much for taking the time to review this manuscript. 

Round 2

Reviewer 1 Report (Previous Reviewer 2)

Comments and Suggestions for Authors

It can be published

Comments on the Quality of English Language

Acceptable

This manuscript is a resubmission of an earlier submission. The following is a list of the peer review reports and author responses from that submission.

Round 1

Reviewer 1 Report

Comments and Suggestions for Authors

The study is interesting. Limited data presentation decreases the novelty of the study. the current study did not provide journal scopes.

1- Please, abbreviations in the first place have to be written in full names.

2- Please add more references (lines 49-55) to support your hypothesis about BACE1 as an early detectable marker.

3- Please add the date to line 163.

4- why you did not compare CTRL and FTD groups?

5- The data is too limited and we can not judge the early detecting biomarker evaluating only by one factor.

6- Language needs grammar corrections.

7- There are some articles that have the same hypothesis:

https://doi.org/10.1186/s12916-024-03418-8; DOI:10.1186/s13195-020-00686-3.

Comments on the Quality of English Language

Language needs grammar corrections.

is, are some s are missing in the text. 

Reviewer 2 Report

Comments and Suggestions for Authors

The most important issue with this paper:

The manuscript predominantly employs univariate analysis with adjustments for age and gender using Generalized Linear Models. While this approach provides some insights, it does not fully account for the potential interrelationships and dependencies between multiple biomarkers, which could be crucial for a more comprehensive understanding of the differential diagnosis between Alzheimer's Disease (AD) and Frontotemporal Dementia (FTD). It is necessary for a paper like this with Sample Size. Without multivariate analysis, the reliability of these findingsis highly questionable.

Comments

1. In the first sentences of the Introduction section, please add some statistics about the prevalence or mortality rate of AD and FTD to highlight the importance and necessity of your project for the audience.

2. The study uses a sample size of only 30 FTD patients and 31 AD patients, which is insufficient to draw meaningful conclusions. Larger, more diverse cohorts are needed to ensure the findings are statistically significant and generalizable.

3. The manuscript does not provide novel insights into the use of BACE1 as a biomarker for differentiating between AD and FTD. Similar studies have already been conducted and reported in the literature, reducing the impact and originality of the current study. PLEASE CLARIFY you novelty clearly.

4. The methodological approach lacks rigor. The study relies heavily on a single assay type without cross-validation using different techniques or additional biomarkers that could corroborate the findings. The details of your current statistical analysis is not even totally clear. The data presentation in the manuscript is inconsistent and lacks clarity. For instance, the statistical analysis details are insufficiently described, and the figures do not adequately support the text's claims. The statistical analysis appears inadequate. The authors do not provide a detailed explanation of the statistical methods used, and there is no mention of adjustments for multiple comparisons, which could lead to type I errors.

5. The authors make broad claims about the potential of serum BACE1 activity as a differential diagnostic tool for AD and FTD without sufficient evidence to support these claims. The conclusions are not justified based on the data presented.

6. The manuscript fails to adequately discuss the study's limitations, such as potential confounding factors, and the need for further validation in larger, independent cohorts. And also future trends: AD is a time-dependence disorders and recent studies have highlighted the significant impact of brain characteristics, such as intracranial compliance and brain creep, on the treatment of brain disorders like AD and hydrocephalus. These factors should be suggested and considered for their impact on the results in addition to the current biomarkers for future studies.

7. The study does not sufficiently validate BACE1 as a biomarker. There is no comparison with other established biomarkers, and the sensitivity and specificity of BACE1 in differentiating AD from FTD are not robustly demonstrated.

8. The manuscript lacks detailed information on ethical considerations and the process for obtaining informed consent from participants, which is crucial for studies involving human subjects.

9. The methods section does not provide enough detail to ensure the study can be replicated. Key experimental details and protocols are missing, making it difficult for other researchers to reproduce the findings.

10. The study does not include longitudinal data to assess how serum BACE1 activity changes over time in AD and FTD patients, which is critical for establishing its utility as a biomarker.

Comments on the Quality of English Language

Medium level

Reviewer 3 Report

Comments and Suggestions for Authors

In this work, the authors analyzed serum BACE1 activity from the Alzheimer’s disease (AD) and frontotemporal dementia (FTD) patients and compared with those obtained from the healthy individuals. They observed a significant increase in serum BACE1 activity in AD when compared to FTD and control samples. This indicates that serum BACE1 activity may be a potential biomarker that can be used for definite diagnosis of the two diseases. However, there are a couple of concerns that should be addressed prior to publication.

1.      In Fig 1, BACE1 level in FTD appears lower than control, despite no statistically significant difference; however, in Fig 2, BACE1 level in FTD appears higher than the control. Are the two control groups included in Fig 1 and 2 the same or different. If different, authors need to provide rationale. Additionally, their Figure Legend should indicate N value in each figure to confirm whether any sample was excluded from the analysis.

 2.      The manuscript contains some grammatical errors that need corrections.

Comments on the Quality of English Language

The manuscript contains some grammatical errors that need corrections.